Nocturnal substrate association of four coral reef fish groups (parrotfishes, surgeonfishes, groupers and butterflyfishes) in relation to substrate architectural characteristics

Nanami Atsushi nanami_atsushi71@fra.go.jp
Environment and Fisheries Applied Techniques Research Department, Fisheries Technology Institute, Japan Fisheries Research and Education Agency, Yaeyama Field Station, Coastal and Inland Fisheries Ecosystem Division , Ishigaki , Okinawa , Japan
Johnson Magnus
Electronic publication date: 2024 Jul 19
Publication date: 2024
Volume: 12
Electronic Location ID: e17772
Received 2024 Jan 16; Accepted 2024 Jun 28
Copyright: ©2024 Nanami
Copyright year: 2024
Copyright holder: Nanami
License: This is an open access article distributed under the terms of the Creative Commons Attribution License, which permits unrestricted use, distribution, reproduction and adaptation in any medium and for any purpose provided that it is properly attributed. For attribution, the original author(s), title, publication source (PeerJ) and either DOI or URL of the article must be cited.
License URL: https://creativecommons.org/licenses/by/4.0/

Keywords: Nocturnal substrate association, Parrotfishes, Surgeonfishes, Groupers, Butterflyfishes, Substrate characteristics, Sleeping site

Funding: The authors received no funding for this work.

==============================
Although numerous coral reef fish species utilize substrates with high structural complexities as habitats and refuge spaces, quantitative analysis of nocturnal fish substrate associations has not been sufficiently examined yet. The aims of the present study were to clarify the nocturnal substrate associations of 17 coral reef fish species (nine parrotfish, two surgeonfish, two grouper and four butterflyfish) in relation to substrate architectural characteristics. Substrate architectural characteristics were categorized into seven types: (1) eave-like space, (2) large inter-branch space, (3) overhang by protrusion of fine branching structure, (4) overhang by coarse structure, (5) uneven structure without large space or overhang, (6) flat and (7) macroalgae. Overall, fishes were primarily associated with three architectural characteristics (eave-like space, large inter-branch space and overhang by coarse structure). The main providers of these three architectural characteristics were tabular and corymbose Acropora, staghorn Acropora, and rock. Species-specific significant positive associations with particular architectural characteristics were found as follows. For the nine parrotfish species, Chlorurus microrhinos with large inter-branch space and overhang by coarse structure; Ch. spilurus with eave-like space and large inter-branch space; Hipposcarus longiceps with large inter-branch space; Scarus ghobban with overhang by coarse structure; five species (Scarus forsteni, S. niger, S. oviceps, S. rivulatus and S. schlegeli) with eave-like space. For the two surgeonfish species, Naso unicornis with overhang by coarse structure; N. lituratus with eave-like space. For the two grouper species, Plectropomus leopardus with eave-like space; Epinephelus ongus with overhang by coarse structure. For the four butterflyfish species, Chaetodon trifascialis with eave-like space and large inter-branch space; C. lunulatus and C. ephippium with large inter-branch space; C. auriga showed no significant associations with any architectural characteristics. Four species (Ch. microrhinos, H. longiceps, S. niger and N. unicornis) also showed clear variations in substrate associations among the different fish size classes. Since parrotfishes, surgeonfishes and groupers are main fisheries targets in coral reefs, conservation and restoration of coral species that provide eave-like space (tabular and corymbose Acropora) and large inter-branch space (staghorn Acropora) as well as hard substrates with coarse structure that provide overhang (rock) should be considered for effective fisheries management in coral reefs. For butterflyfishes, coral species that provide eave-like space (tabular Acropora) and large inter-branch space (staghorn Acropora) should also be conserved and restored for provision of sleeping sites.

Introduction

Coral reefs provide various substrates with high structural complexities, which are key determinants supporting high species diversity of marine organisms (Jaap, 2000; Yanovski, Nelson & Abelson, 2017). Numerous coral reef fish species utilize substrates with a high structural complexity as habitats and refuge spaces (Luckhurst & Luckhurst, 1978; Ménard et al., 2012; Richardson et al., 2017; Oren et al., 2023). Species-specific habitat associations to specific substrates or structural complexities have also been reported (Wilson et al., 2008; Ticzon et al., 2012; Untersteggaber, Mitteroecker & Herler, 2014; Nanami, 2023). Such species-specific habitat associations have been shown to influence populations through survivorship (Fakan et al., 2024).

Coral reef fishes provide various ecosystem services such as natural food production, ornamental resources, aquarium resources, habitat maintenance and recreation (Moberg & Folke, 1999; Laurans et al., 2013; Elliff & Kikuchi, 2017; Sato et al., 2020). This diverse ecosystem services provided by coral reefs include supporting (biodiversity benefit and habitat), regulating (coastal protection and water quality), provisioning (fishery and materials) and cultural services (Woodhead et al., 2019). Among the diverse ecosystem services, the provision of fisheries targets is recognized as an essential service (Elliff & Kikuchi, 2017; Woodhead et al., 2019). Specifically, parrotfishes (family Labridae: Scarini), groupers (family Epinephelidae) and surgeonfishes (family Acanthuridae) are the main targets of commercial fisheries in many countries in tropical and sub-tropical regions (e.g., Bejarano et al., 2013; Taylor et al., 2014; Akita et al., 2016; Frisch et al., 2016). Provision of ornamental resources or aquarium resources is also an important ecosystem service in coral reefs, and butterflyfishes (family Chaetodontidae) are regarded as a target in the aquarium trade for their popularity as ornamental fishes (Tissot & Hallacher, 2003; Wabnitz et al., 2003; Lawton, Pratchett & Delbeek, 2013).

Several studies have revealed species-specific spatial distributions of these four fish groups in relation to topographic features or environmental characteristics (e.g., Newman, Williams & Russ, 1997; Hoey & Bellwood, 2008; Hernández-Landa et al., 2014; Nanami, 2020; Nanami, 2021). Previous studies have also revealed the foraging substrates for parrotfishes (Bonaldo & Rotjan, 2018; Nicholson & Clements, 2020), surgeonfishes (Robertson & Gaines, 1986), groupers (Wen et al., 2013a) and butterflyfishes (Cole & Pratchett, 2013; Pratchett, 2013). In contrast, precise substrate characteristics (e.g., coral species, coral morphology and physical structure) that were directly associated by fish individuals of these fish groups have not been sufficiently examined. This is because most individuals belonging to these fish groups are diurnally active and rarely show hiding behavior with specific substrates. Although some previous studies have revealed the diurnal substrate associations of groupers (Nanami et al., 2013; Wen et al., 2013b), their nocturnal associations have not been examined yet.

Several previous studies have shown high site fidelity by parrotfishes (Welsh & Bellwood, 2012; Pickholtz et al., 2022), surgeonfishes (Meyer & Holland, 2005; Marshell et al., 2011), groupers (Zeller, 1997; Matley, Heupel & Simpfendorfer, 2015; Nanami et al., 2018) and butterflyfishes (Yabuta & Berumen, 2013). For instance, Pickholtz et al. (2022) revealed that three parrotfish species repetitively used specific spaces during nocturnal periods in the Red Sea. Marshell et al. (2011) showed high site fidelity during nocturnal periods by two surgeonfish species in Guam. From the results of these studies, nocturnal substrate associations might be observed due to their nocturnal high site fidelity.

Improving the understanding of nocturnal substrate associations of fishes would provide useful ecological information for effective ecosystem management such as habitat protection and restoration by implementation of marine protected areas. This is because conservation of critical habitats for target species is crucial for marine protected area planning (Kelleher, 1999; Green, White & Kilarski, 2013). Thus, nocturnal substrate association of fishes should be determined to understand better the critical habitats in terms of fish nocturnal habitat utilization. In addition, parrotfishes, groupers and surgeonfishes are primary target species in the Pacific Islands fishery and nighttime spear fishing is one of the methods to catch inactive individuals (Gillett & Moy, 2006). Thus, identifying the substrate characteristics that are utilized by fishes as sleeping sites is critical for conservation of fishing points. Although some previous studies have revealed nocturnal fish substrate associations (Hobson, 1965; Robertson & Sheldon, 1979; Pickholtz et al., 2023), quantitative analysis of nocturnal substrate associations in relation to substrate availability has not been sufficiently examined yet.

The aims of the present study were to understand the nocturnal substrate associations of four coral reef fish groups (parrotfishes, surgeonfishes, groupers and butterflyfishes), which provide many ecosystem services in coral reefs. Specifically, the aims were to understand nocturnal substrate associations of fish in terms of (1) architectural characteristics (physical structure) and (2) more precise aspects (coral morphology, live coral or dead coral, and non-coralline substrates). The results will enable a more comprehensive understanding of the association between coral reef fishes and substrate characteristics, and may be useful in helping us to anticipate changes in fish assemblages structure that may occur due to anthropogenically or climate induced changes in coral reefs.

Materials and Methods

The study was conducted by field observations. Fish individuals that were caught for sampling by spear were euthanized immediately to minimize suffering. Okinawa prefectural government fisheries coordination regulation No. 37 approved the sampling procedure (https://www.pref.okinawa.jp/_res/projects/default_project/_page_/001/011/218/r02kisoku.pdf), which permits capture of marine fishes on Okinawan coral reefs for scientific purposes.

Fish survey and study species

This study was conducted at Sekisei lagoon and Nagura Bay in the Yaeyama Islands, Okinawa, Japan (Fig. 1). Nocturnal underwater observations (1830 h–23:00 h) were conducted at 19 sites between November 2021 and March 2022. Using SCUBA and flashlights, the first diver swam in a zigzag pattern and searched for inactive individuals that were associated with substrates (Fig. 2), taking special care not to overlap with previous courses. The second diver followed the first diver with a data collection sheet. When the first diver found a focal fish, the second diver recorded the species, total length (TL) and substrate with which the focal fish individual was associated. In some cases, the whole body of the fish was not completely observed due to hiding behavior within the substrate. In this case, the focal fish individual was collected by spear and the TL was measured. Over 40 min observations were conducted at each site (ranging from 40 to 72 min, average μ = 52.3 ± 9.2 s.d. minutes). According to Nanami (2021), average distance of 1-minute swimming was 17.4 m. Thus, the estimated distance of each time survey was 17.4 m × survey minutes. Since the width of the time transect was 5 m, the estimated area was distance ×5 m2.

Figure 1 Maps showing the location of the Yaeyama Islands (A), study area (B) and the 19 study sites used for examining nocturnal substrate associations of fishes (C).

(A) Map created by processing Geospatial Information Authority (https://mapps.gsi.go.jp/maplibSearch.do#1). The aerial photographs in (B) and (C) were provided by the International Coral Reef Research and Monitoring Center.

Figure 2 (A–Q) Examples of inactive fish individuals that were associated with substrates at nighttime for the 17 species.

One example is shown for each species. For more details about substrate associations of fishes, see Figs. 4–9. All fish photographs were taken by the author (A. Nanami).

During the observation period, 19, two, nine, and 12 parrotfish, surgeonfish, grouper and butterflyfish species were identified, respectively (Table 1). Among them, nine, two, two, and four species showed higher frequencies (total number of individuals was 10 or more) for parrotfishes, surgeonfishes, groupers and butterflyfishes, respectively. Thus, the data analysis was conducted in two steps. The first step was to clarify the species-level substrate associations by using above-mentioned 17 most frequent species (nine, two, two and four species for parrotfishes, surgeonfishes, groupers and butterflyfishes, respectively). The second step was to clarify the family-level substrate associations by using all species including both frequent and less-frequent species (19, two, nine, and 12 species for parrotfishes, surgeonfishes, groupers and butterflyfishes, respectively).

Table 1 List and number of individuals of fishes belong to four fish groups (parrotfishes, surgeonfishes, groupers and butterflyfishes) that were observed for nocturnal substrate association.

Family	Species	Number of individuals	Size range (TL: cm)	Analysis	Substrate architectural characteristics		
					Eave-like	Large Inter-branch	Overhang by fine branching	Overhang by coarse structure	Uneven	Flat	Macroalgae	
Parrotfishes (Labridae: Scarini)	Cetoscarus bicolor	3	44–46		2			1				
	Chlorurus bowersi	5	28–33		2	2	1					
	Chlorurus japanensis	1	33		1							
	Chlorurus microrhinos	24	25–62	X	2.5*	8		13.5*				
	Chlorurus spilurus	45	20–32	X	20	14	9	2				
	Hipposcarus longiceps	22	15–53	X	3	10	1	8				
	Scarus chameleon	1	25			1						
	Scarus festivus	2	27–40				2					
	Scarus forsteni	15	20–40	X	10			5				
	Scarus frenatus	3	23–33		2			1				
	Scarus ghobban	21	24–57	X	5.5*	1	1	13.5*				
	Scarus hypselopterus	6	25–27		4		1	1				
	Scarus niger	11	20–35	X	6.5*	2.5*		2				
	Scarus oviceps	14	20–34	X	13			1				
	Scarus prasiognathos	1	35		1							
	Scarus quoyi	1	25		1							
	Scarus rivulatus	23	25–35	X	19	1	2	1				
	Scarus schlegeli	22	18–29	X	16.5*		1	4.5*				
	Scarus spinus	5	24–25		2			3				
Surgeonfishes (Acanthuridae)	Naso lituratus	23	15–30	X	11		3	9				
	Naso unicornis	32	30–70	X	5		1	26				
Groupers (Epinephelidae)	Cephalopholis argus	1	28					1				
	Cephalopholis miniata	2	23–24		1			1				
	Epinephelus fuscoguttatus	1	59					1				
	Epinephelus hexagonatus	1	31					1				
	Epinephelus ongus	106	10–32	X	15	16	30	45				
	Epinephelus polyphekadion	3	25–40		1	2						
	Epinephelus tauvina	2	29–37					2	1	
Plectropomus leopardus	30	20–62	X	12	2	5	10				
	Variola louti	2	35–47		1		1					
Butterflyfishes (Chaetodontidae)	Chaetodon auriga	16	12–20	X	5	2	2	7				
	Chaetodon auripes	1	13				1					
	Chaetodon baronessa	5	13–15		1	2		2				
	Chaetodon bennetti	2	8–16				1	1				
	Chaetodon ephippium	10	13–18	X		5		5				
	Chaetodon lunulatus	61	6–14	X	6	43	9	3				
	Chaetodon ornatissimus	6	13–17		1			5				
	Chaetodon plebeius	2	8–12		1	1						
	Chaetodon trifascialis	21	5–13	X	9	9	2	1				
	Chaetodon ulietensis	2	10–12			1		1				
	Chaetodon vagabundus	8	10–15				1	7				
	Forcipiger flavissimus	1	15					1				
Notes.

X, fish species that were selected for analyses (total number of individuals were 10 individuals and over).

* since one individual utilized two categories of substrates (the two substrates were closely located to each other and one focal fish individual was associated with both substrates simultaneously), 0.5 individuals were assigned for each substrate as substrate association.

Table 2 Relationship between seven categories of substrate architectural characteristics (physical structure) and 25 substrate types.

Substrate architectural characteristics	Substrate	
Eave-like space	Corymbose Acropora	
	Tabular Acropora	
	Foliose coral	
	Dead corymbose Acropora	
	Dead tabular Acropora	
	Dead foliose coral	
Large inter-branch space	Staghorn Acropora	
	Dead staghorn Acropora	
Overhang by fine branching structure	Branching Acropora	
	Bottlebrush Acropora	
	Non-acroporid branching coral	
	Pocillopora	
	Dead branching Acropora	
	Dead bottlebrush Acropora	
	Dead non-acroporid branching coral	
	Dead Pocillopora	
Overhang by coarse structure	Massive coral	
	Dead massive coral	
	Rock	
Uneven structure without large space or overhang	Other coral	
	Dead other coral	
	Soft coral	
Flat	Coral rubble	
	Sand	
Macroalgae	Macroalgae	

Data collection of substrate availability

Substrate availability at the 19 study sites was recorded during daytime. The locations of sites where nocturnal observations were conducted were recorded using a portable GPS receiver (GARMIN GPSMAP 64csx). Then, video recordings were used to record substrates on the seafloor during 20 min at each site. Static images were extracted at 10-second intervals by QuickTime Player software (version 7.6), yielding 121 static images for each site. For each image, the substrate at the center of the static image was recorded.

Substrate categorization and definition of substrate architectural characteristics

Substrates were categorized into 25 types and substrate architectural characteristics (physical structure) were categorized into seven types with some modification from several previous studies (Gardiner & Jones, 2005; Wilson et al., 2008; Nanami, 2020; Doll et al., 2021) as follows (Table 2, Fig. 3): (1) eave-like space, (2) large inter-branch space, (3) overhang provided by protrusion of fine branching structure, (4) overhang by coarse structure, (5) uneven structure without large space or overhang, (6) flat and (7) macroalgae.

Figure 3 (A–G) Schematic diagrams of the seven types of substrate architectural characteristics (physical structure) and some examples of substrates for each type.

Light green areas represent spaces that are potentially utilized by fishes as sleeping site. For more details about relationships between structural characteristics and substrates, see Table 2. All substrate photographs were taken by the author (A. Nanami).

Figure 4 (A–C) Relative frequency (%) of fish individuals associated with substrates and substrate availability for the three parrotfish species (Chlorurus microrhinos,C. spilurus and Hipposcarus longiceps).

Left and right figures represent results using the seven types of substrate architectural characteristics (physical structure) and 19 substrate types, respectively. Numbers adjacent to bars represent the number of individuals that were associated with the focal substrate. For right figures, data from 19 substrate types among 25 the substrate types are shown, since no fish individuals were associated with the remaining six substrate types (other coral, dead other coral, soft coral, coral rubble, sand and macroalgae). An asterisk indicates that since one individual utilized two categories of substrates (the two substrates were closely located to each other and one focal fish individual was associated with both substrates simultaneously), 0.5 individuals were assigned for each substrate as substrate association. All fish photographs were taken by the author (A. Nanami).

Data analysis for substrate association

The analyses were conducted in two steps. The first step was to clarify the associations between fish species and the seven types of substrate architectural characteristics (physical structure). The second step was to clarify the associations between fish species and the 25 substrate types.

Fish associations were analyzed by using “resource selection ratio” (Manly et al., 2002). The approach follows previous studies that have applied this index to examine the quantitative degree of substrate association of coral reef fishes to specific substrate characteristics (e.g., Gardiner & Jones, 2005; Wilson et al., 2008; Doll et al., 2021; Nanami, 2023). This index also shows 95% confidence intervals, which can be used to test the statistical significance of the substrate association of fishes for each substrate type.

The resource selection ratio was calculated as: wi=oi/πi

where wi is the resource selection probability function, oi is the proportion of the ith substrate that was used by a focal fish species, and πi is the proportion of the ith substrate that was available in the study area (Manly et al., 2002). For multiple comparisons, Bonferroni Z corrections were used in order to calculate the 95% confidence interval (CI) for each wi. The formula used to calculate the 95% CI was: 95%CI=Za/2k√oi1−oi/U+πi2

where Za/2k is the critical value of the standard normal distribution corresponding to an upper tail area of a/2k, a is 0.05, k is the number of substrate categories, and U+ is the total number of individuals of the focal fish species. Substrates with wi ±95% CI above and below 1 indicate a significantly positive and negative association, respectively. Substrates with wi ±95% CI encompassing 1 had no significant positive or negative association.

In addition, the standardized selection ratio that indicates relative degree among substrates for habitat selection was calculated as follows: Bi=wi ∑wi.

If a focal species shows Bi and Bj for i th and jth substrates, ith substrate is selected with Bi/Bj times the probability of jth substrate.

Both species level (17 species) and family level (four families) data analyses were performed.

Variations in substrate associations among different fish size classes

To investigate the variations in substrate associations among different fish size classes, fish individuals were divided into three size classes as follows: (1) TL ≤ 29 cm (smaller-sized); (2) TL = 30–39 cm (medium-sized) and (3) TL ≥ 40 cm (larger-sized). Then, their degree of substrate association was analyzed. Five species (Scarus schlegeli, Chaetodon trifascialis, C. lunulatus, C. ephippium and C. auriga) were excluded from the analysis, since total length of the all individuals were 29 cm or less for the five species.

Data preparation prior to analysis

All data for substrate associations by fish were obtained from the 19 study sites were pooled for the analysis. Although all data for substrate availability from the 19 sites were also pooled for the analysis, a modification was applied due to the difference in observation time among the 19 sites (see substrate availability raw data; Supplemental Information). Namely, substrate compositions at sites with longer fish observation durations should be included with greater proportions whereas substrate compositions at sites with shorter time observation durations should be included with lower proportions. The degree of the proportion was assigned by the observation duration at the site. Thus, the modification was as follows: Overall proportion ofith substrate= ∑j=119PijTj ∑i=1k ∑j=119PijTj

where Pij is the proportion of ith substrate at site j, Tj is the observation duration (minutes) at site j, and k is the number of substrate types (k = 7 for seven types of substrate architectural structure and k = 25 for twenty-five substrate types).

Overall trend in substrate association

To summarize species-specific differences in substrate association, a principal component analysis (PCA) and cluster analysis using the group average linkage method with the Bray–Curtis similarity index was applied based on the number of fishes by including data from the seventeen fish species. Analyses were performed using PRIMER (version 6) software package (Clarke & Warwick, 1994). For plotting the PCA score of each fish species, data about nocturnal substrate association were also shown by pie charts. Additional PCA was performed to clarify the variations in substrate associations among the above-mentioned three fish size classes.

Results

Parrotfishes

Chlorurus microrhinos was primarily associated with large inter-branch space (staghorn Acropora) and overhang by coarse structure (rock) (Fig. 4A). Significant positive associations with large inter-branch space and overhang by coarse structure were found (Table 3, Table S1). However, no significant substrate associations were found for any types of 25 substrates (Table 4, Table S2). For size difference, smaller-sized and medium-sized individuals were primarily associated with large inter-branch space (staghorn Acropora), whereas larger-sized individuals were primarily associated with overhang by coarse structure (rock: Fig. S1).

Table 3 Results of statistical significance of substrate association of the nine parrotfish species calculated by resource selection ratio for seven types of substrate architectural characteristics.

Substrate architectural characteristics	Chlorurus microrhinos	Chlorurus spilurus	Hipposcarus longiceps	Scarus ghobban	Scarus forsteni	Scarus niger	Scarus oviceps	Scarus rivulatus	Scarus schlegeli	
Eave-like	N.S.	Positive	N.S.	N.S.	Positive	Positive	Positive	Positive	Positive	
Large inter-branch	Positive	Positive	Positive	N.S.	–	N.S.	–	N.S.	–	
Overhang by fine branching	–	N.S.	Negative	Negative	N.S.	–	–	N.S.	Negative	
Overhang by coarse strure	Positive	Negative	N.S.	Positive	–	N.S.	N.S.	Negative	N.S.	
Uneven	–	–	–	–	–	–	–	–	–	
Flat	–	–	–	–	–	–	–	–	–	
Macroalge	–	–	–	–	–	–	–	–	–	
Notes.

Significant positive associations are shown as bold characters.

N.S non significant associations

– No fishes were found on the substrates

Table 4 Results of substrate association of the nine parrotfish species calculated by resource selection ratio for 25 substrate types.

Substrate architectural characteristics	Substrate type	Chlorurus microrhinos	Chlorurus spilurus	Hipposcarus longiceps	Scarus ghobban	Scarus forsteni	Scarus niger	Scarus oviceps	Scarus rivulatus	Scarus schlegeli	
Eave-like	Corymbose Acropora	N.S.	N.S.	N.S.	N.S.	N.S.	N.S.	N.S.	N.S.	Positive	
	Tabular Acropora	–	N.S.	N.S.	N.S.	Positive	N.S.	Positive	Positive	N.S.	
	Foliose coral	–	–	–	–	–	N.S.	–	–	N.S.	
	Dead corymbose Acropora	N.S.	N.S.	–	–	–	–	–	–	–	
	Dead tabular Acropora	N.S.	N.S.	N.S.	N.S.	N.S.	N.S.	–	N.S.	N.S.	
	Dead foliose coral	–	–	–	–	–	–	–	–	–	
Large Inter-branch	Staghorn Acropora	N.S.	Positive	Positive	N.S.	–	N.S.	–	N.S.	–	
	Dead staghorn Acropora	–	–	–	–	–	–	–	–	–	
Overhang by fine branching	Branching Acropora	–	–	–	–	–	–	–	–	–	
	Bottlebrush Acropora	–	N.S.	–	–	–	–	–	–	–	
	Non-acroporid branching coral	–	N.S.	N.S.	N.S.	–	–	–	N.S.	N.S.	
	Pocillopora	–	–	–	–	–	–	–	–	–	
	Dead branching Acropora	–	–	–	–	–	–	–	–	–	
	Dead bottlebruch Acropora	–	–	–	–	–	–	–	–	–	
	Dead non-acroporid branching coral	–	N.S.	–	–	–	–	–	–	–	
	Dead Pocillopora	–	–	–	–	–	–	–	–	–	
Overhang by coarse structure	Massive coral	N.S.	–	–	N.S.	–	–	–	–	N.S.	
	Dead massive coral	–	–	–	N.S.	–	–	–	–	–	
	Rock	N.S.	Negative	N.S.	N.S.	N.S.	N.S.	N.S.	Negative	N.S.	
Uneven	Other coral	–	–	–	–	–	–	–	–	–	
	Dead other coral	–	–	–	–	–	–	–	–	–	
	Soft coral	–	–	–	–	–	–	–	–	–	
Flat	Coral rubble	–	–	–	–	–	–	–	–	–	
	Sand	–	–	–	–	–	–	–	–	–	
Macroalgae	Macroalgae	–	–	–	–	–	–	–	–	–	
Notes.

Significant positive associations are shown as bold characters.

N.S. non significant associations

– no fishes were found on the substrates

Chlorurus spilurus was primarily associated with eave-like space (corymbose Acropora and tabular Acropora), large inter-branch space (staghorn Acropora) and overhang by fine branching structure (non-acroporid branching coral) (Fig. 4B). Significant positive associations with eave-like space and large inter-branch space were found (Table 3, Table S1). For eave-like space, no significant substrate-specific associations were found (Table 4, Table S2). For large inter-branch space, significant positive association with staghorn Acropora was found (Table 4, Table S2). In contrast, a significant negative association with overhang by coarse structure (rock) was found (Tables 3 and 4, Tables S1, S2). By size, smaller- and medium-sized individuals showed relatively greater proportion of association with eave-like space (corymbose and tabular Acropora) and large inter-branch space (staghorn Acropora), respectively (Fig. S2).

Hipposcarus longiceps was primarily associated with large inter-branch space (staghorn Acropora) and overhang by coarse structure (rock) (Fig. 4C). Significant positive and negative associations with large inter-branch space (staghorn Acropora) and overhang by fine branching structure were found, respectively (Tables 3 and 4, Tables S1, S2). By size, smaller-, medium- and larger-sized individuals showed relatively greater proportion of association with large inter-branch space (staghorn Acropora), overhang by coarse structure (rock) and eave-like space (tabular and dead tabular Acropora), respectively (Fig. S3).

Scarus ghobban was primarily associated with eave-like space (corymbose Acropora) and overhang by coarse structure (massive coral and rock) (Fig. 5A). Although this species showed respectively significant positive and negative associations with overhang by coarse structure and overhang by fine branching structure (Table 3, Table S1), no significant substrate-specific associations were found (Table 4, Table S2). All three size classes showed relatively greater proportion of association with overhang by coarse structure (massive coral and rock: Fig. S4).

Figure 5 (A–C) Relative frequency (%) of fish individuals associated with substrates and substrate availability for the three parrotfish species Scarus ghobban, S. forsteni and S. niger).

Left and right figures represent results using the seven types of substrate architectural characteristics (physical structure) and 19 substrate types, respectively. Numbers adjacent to bars represent the number of individuals that were associated with the focal substrate. For right figures, data from 19 substrate types among 25 the substrate types are shown, since no fish individuals were associated with the remaining six substrate types (other coral, dead other coral, soft coral, coral rubble, sand and macroalgae). An asterisk (*) indicates that since one individual utilized two categories of substrates (the two substrates were closely located to each other and one focal fish individual was associated with both substrates simultaneously), 0.5 individuals were assigned for each substrate as substrate association. All fish photographs were taken by the author (A. Nanami).

Five species (Scarus forsteni, S. niger, S. oviceps, S. rivulatus and S. schlegeli) were primarily associated with eave-like space (corymbose Acropora and tabular Acropora) (Figs. 5B, 5C, 6A–6C) and showed a significant positive association with the eave-like space (Table 3, Table S1). Three species (S. forsteni, S. oviceps and S. rivulatus) and one species (S. schlegeli) showed positive associations with tabular Acropora and corymbose Acropora, respectively (Table 4, Table S2). In contrast, S. niger did not show any substrate-specific associations (Table 4, Table S2). For size difference, two size classes (smaller- and larger-sized) individuals of S. forsteni showed greater proportion in association with eave-like space (tabular Acropora) while medium-sized fish were associated with overhang by coarse structure (rock) , respectively (Fig. S5). Smaller-sized individuals of S. niger showed greater proportion in association with eave-like space (mainly tabular Acropora) and medium-sized with large inter-branch space (staghorn Acropora), respectively (Fig. S6). In contrast, all size classes of the two species (S. oviceps and S. rivulatus) were primarily associated with eave-like space (mainly tabular Acropora: Figs. S7, S8).

Figure 6 (A–C) Relative frequency (%) of fish individuals associated with substrates and substrate availability for the three parrotfish species Scarus oviceps, S. rivulatus and S. schlegeli).

Left and right figures represent results using the seven types of substrate architectural characteristics (physical structure) and 19 substrate types, respectively. Numbers adjacent to bars represent the number of individuals that were associated with the focal substrate. For right figures, data from 19 substrate types among 25 the substrate types are shown, since no fish individuals were associated with the remaining six substrate types (other coral, dead other coral, soft coral, coral rubble, sand and macroalgae). An asterisk (*) indicates that since one individual utilized two categories of substrates (the two substrates were closely located to each other and one focal fish individual was associated with both substrates simultaneously), 0.5 individuals were assigned for each substrate as substrate association. All fish photographs were taken by the author (A. Nanami).

Surgeonfishes

Naso unicornis was primarily associated with overhang by coarse structure (rock: Fig. 7A) and showed a significant positive association with the substrate (Tables 5 and 6, Tables S3, S4). A significant negative association with overhang by fine branching structure was also found (Table 5, Table S3). By size, smaller- and larger-sized individuals were primarily associated with eave-like space (dead tabular Acropora) and overhang by coarse structure (rock), respectively (Figs. S9A, S9C). Medium-sized individuals was associated with both eave-like space (tabular Acropora) and overhang by coarse structure (rock: Fig. S9B).

Figure 7 (A–B) Relative frequency (%) of fish individuals associated with substrates and substrate availability for the two surgeonfish species.

Left and right figures represent results using the seven types of substrate architectural characteristics and 19 substrates types, respectively. Numbers adjacent to bars represent the number of individuals that were associated with the focal substrate. For right figures, data from 19 substrate types among the 25 substrate types were shown, since no fish individuals were associated with the remaining six substrate types (other coral, dead other coral, soft coral, coral rubble, sand and macroalgae). All fish photographs were taken by the author (A. Nanami).

Table 5 Results of statistical significance of substrate association of the two surgeonfish, two grouper and four butterflyfish species calculated by resource selection ratio for seven types of substrate architectural characteristics.

Substrate architectural characteristics	Naso unicornis	Naso lituratus	Plectropomus leopardus	Epinephelus ongus	Chaetodon trifascialis	Chaetodon lunulatus	Chaetodon ephippium	Chaetodon auriga	
Eave-like	N.S.	Positive	Positive	N.S.	Positive	N.S.	–	N.S.	
Large inter-branch	–	–	N.S.	N.S.	Positive	Positive	Positive	N.S.	
Overhang by fine branching	Negative	N.S.	N.S.	N.S.	N.S.	N.S.	–	N.S.	
Overhang by coarse structure	Positive	N.S.	N.S.	Positive	Negative	Negative	N.S.	N.S.	
Uneven	–	–	–	–	–	–	–	–	
Flat	–	–	Negative	–	–	–	–	–	
Macroalge	–	–	–	–	–	–	–	–	
Notes.

Significant positive associations are shown as bold characters.

N.S. non significant associations

– no fishes were found on the substrates

Table 6 Results of statistical significance of substrate association of the two surgeonfish, two grouper and four butterflyfish species calculated by resource selection ratio for 25 substrate types.

Substrate architectural characteristics	Substrate type	Naso unicornis	Naso lituratus	Plectropomus leopardus	Epinephelus ongus	Chaetodon trifascialis	Chaetodon lunulatus	Chaetodon ephippium	Chaetodon auriga	
Eave-like	Corymbose Acropora	–	N.S.	N.S.	N.S.	–	N.S.	–	N.S.	
	Tabular Acropora	N.S.	N.S.	N.S.	N.S.	Positive	N.S.	–	–	
	Foliose coral	–	–	–	–	–	–	–	–	
	Dead corymbose Acropora	–	–	N.S.	–	–	–	–	–	
	Dead tabular Acropora	N.S.	–	N.S.	N.S.	–	–	–	N.S.	
	Dead foliose coral	–	–	–	–	–	–	–	–	
Large inter-branch	Staghorn Acropora	–	–	N.S.	N.S.	Positive	Positive	N.S.	N.S.	
	Dead staghorn Acropora	–	–	–	N.S.	–	–	–	–	
Overhang by fine branching	Branching Acropora	–	–	–	Negative	N.S.	N.S.	–	N.S.	
	Bottlebrush Acropora	–	–	–	N.S.	–	N.S.	–	–	
	Non-acroporid branching coral	N.S.	N.S.	N.S.	Positive	–	N.S.	–	N.S.	
	Pocillopora	–	N.S.	N.S.	N.S.	–	N.S.	–	–	
	Dead branching Acropora	–	–	–	N.S.	–	–	–	–	
	Dead bottlebruch Acropora	–	–	–	–	–	–	–	–	
	Dead non-acroporid branching coral	–	–	N.S.	N.S.	–	–	–	–	
	Dead Pocillopora	–	N.S.	–	–	–	–	–	–	
Overhang by coarse structure	Massive coral	–	N.S.	N.S.	N.S.	–	–	–	N.S.	
	Dead massive coral	N.S.	–	N.S.	–	–	–	N.S.	–	
	Rock	Positive	N.S.	N.S.	N.S.	Negative	Negative	N.S.	N.S.	
Uneven	Other coral	–	–	–	–	–	–	–	–	
	Dead other coral	–	–	–	–	–	–	–	–	
	Soft coral	–	–	–	–	–	–	–	–	
Flat	Coral rubble	–	–	Negative	–	–	–	–	–	
	Sand	–	–	–	–	–	–	–	–	
Macroalge	Macroalgae	–	–	–	–	–	–	–	–	
Notes.

Significant positive associations are shown as bold characters.

N.S. non significant associations

– no fishes were found on the substrates

Naso lituratus was primarily associated with eave-like space (tabular Acropora) and overhang by coarse structure (rock: Fig. 7B). Significant positive association with eave-like space was found (Table 5, Table S3). However, no significant substrate associations were found for any types of 25 substrates (Table 6, Table S4). For size difference, smaller- and medium-sized individuals showed greater proportion in association with eave-like space (mainly tabular Acropora) and overhang by coarse structure (rock), respectively (Fig. S10).

Groupers

Plectropomus leopardus was primarily associated with eave-like space (corymbose and tabular Acropora) and overhang by coarse structure (rock: Fig. 8A). This species showed a significant positive association with eave-like space (Table 5, Table S3), although no significant substrate-specific associations were found (Table 6, Table S4). In contrast, a significant negative association with flat (coral rubble) was found (Tables 5 and 6, Tables S3, S4). By size, medium-sized individuals were primarily associated with eave-like space (mainly corymbose and tabular Acropora: Fig. S11B). However, no clear trends were found for smaller- and larger-sized individuals (Figs. S11A, S11C).

Figure 8 (A–B) Relative frequency (%) of fish individuals associated with substrates and substrate availability for two grouper species.

Left figures represent results using the seven types of substrate architectural characteristics. Right figures represent results using 24 and 19 substrate types for Plectropomus leopardus and Epinephelus ongus, respectively. Numbers adjacent to bars represent the number of individuals that were associated with the focal substrate. For right figures, data from 24 and 19 substrate types among 25 substrate types are shown, since no fish individuals were associated with the remaining one and six substrate types for Plectropomus leopardus (microalgae) and Epinephelus ongus (other coral, dead other coral, soft coral, coral rubble, sand and macroalgae), respectively. All fish photographs were taken by the author (A. Nanami).

Epinephelus ongus was primarily associated with overhang by fine branching structure (non-acroporid branching coral) and overhang by coarse structure (rock: Fig. 8B). A significant positive association with overhang by coarse structure were found (Table 5, Table S3). However, for substrate-specific associations, significant positive and negative associations with non-acroporid branching coral and branching Acropora were respectively found (Table 6, Table S4). All size class individuals showed greater proportions in association with overhang by coarse structure (rock: Fig. S12). Some individuals were also associated with overhang by fine branching structure (branching coral) and this trend was observed for all size classes (Fig. S12).

Figure 9 Relative frequency (%) of fish individuals associated with substrates and substrate availability for the four butterflyfish species.

Left and right figures represent results using the seven types of substrate architectural characteristics and 19 substrate types, respectively. Numbers adjacent to bars represent the number of individuals that were associated with the focal substrate. For right figures, data from 19 substrate types among the 25 substrate types are shown, since no fish individuals were associated with the remaining six substrate types (other coral, dead other coral, soft coral, coral rubble, sand and macroalgae). All fish photographs were taken by the author (A. Nanami).

Figure 10 Results of principal component analysis (PCA) for substrate association of fishes based on five types of substrate architectural characteristics (A, B) and 18 substrates types (C, D).

In A and C, the vectors for two types of architectural characteristics (uneven structure and macroalgae) and seven substrate types (other coral, dead bottlebrush Acropora, dead foliose coral, dead other coral, soft coral, sand and macroalgae) are not shown, since no fish individuals were associated with the substrates. Divisions into multiple groups in (B) and (D) were based on the results of cluster analysis (Fig. S2). Pie charts in (B) and (D) represent proportion of nocturnal substrate association for each fish species. In (B) and (D), fish species names are shown as abbreviations (Ch.mic, Chlorurus microrhinos; Ch.spi: Chlorurus spilurus; H.lon, Hipposcarus longiceps; S.gho, Scarus ghobban; S.for, Scarus forsteni; S.nig, Scarus niger; S.ovi, Scarus oviceps; S.riv, Scarus rivulatus; S.sch, Scarus schlegeli; N.uni, Naso unicornis; N.lit, Naso lituratus; P.leo, Plectropomus leopardus; E.ong, Epinephelus ongus; C.tri, Chaetodon trifascialis; C.lun, Chaetodon lunulatus; C.eph, Chaetodon ephippium; C.arg, Chaetodon auriga). In (D), “Other substrates” includes 11 substrate types (bottlebrush Acropora, non-acroporid branching coral, foliose coral, Pocillopora, dead corymbose Acropora, dead tabular Acropora, dead staghorn Acropora, dead branching Acropora, dead non-acroporid branching coral, dead Pocillopora and coral rubble). For details about data, see Supplemental Information 31 and Supplemental Information 38.

Butterflyfishes

Chaetodon trifascialis was primarily associated with eave-like space (tabular Acropora) and large inter-branch space (staghorn Acropora: Fig. 9A) and showed significant positive associations with these substrates (Tables 5 and 6, Tables S3, S4). This species also showed a significant negative association with overhang by coarse structure (rock: Tables 5 and 6, Tables S3, S4).

Chaetodon lunulatus was primarily associated with large inter-branch space (staghorn Acropora: Fig. 9B) and showed a significant positive association with the substrate (Tables 5 and 6, Tables S3, S4). This species also showed a significant negative association with overhang by coarse structure (rock: Tables 5 and 6, Tables S3, S4).

Chaetodon ephippium was associated with large inter-branch space (staghorn Acropora) and overhang by coarse structure (dead massive coral and rock: Fig. 9C) and showed a significant positive association with large inter-branch space (Table 5, Table S3). However, no significant substrate associations were found for any types of 25 substrate types (Table 6, Table S4).

Chaetodon auriga was primarily associated with eave-like space (corymbose Acropora and dead tabular Acropora) and overhang by coarse structure (rock: Fig. 9D). However, no significant associations with any structural characteristics and substrate types were found (Tables 5 and 6, Tables S3, S4).

Family-level substrate associations

Parrotfishes were primarily associated with eave-like space (corymbose Acropora and tabular Acropora), and some individuals were also associated with large inter-branch space (staghorn Acropora), overhang by fine branching structure (non-acroporid branching coral) and overhang by coarse structure (rock: Fig. S13A). Parrotfishes showed significant positive associations with eave-like space (corymbose Acropora, tabular Acropora and dead tabular Acropora) and large inter-branch space (staghorn Acropora) were found, whereas showed a significant negative association with overhang by fine branching structure (bottlebrush Acropora: Tables S5–S8).

Surgeonfishes were primarily associated with overhang by coarse structure (rock), and some individuals were also associated with eave-like space (tabular Acropora) and overhang by fine branching structure (non-acroporid branching coral) (Fig. S13B). Surgeonfishes showed significant positive associations with eave-like space (tabular Acropora) and overhang by coarse structure (rock: Tables S5–S8). A significant negative association with overhang by fine branching structure was also found (Tables S5, S7).

Groupers were primarily associated with overhang by coarse structure (rock), and some individuals were associated with eave-like space (corymbose Acropora and tabular Acropora), large inter-branch space (staghorn Acropora) and overhang by fine branching structure (non-acroporid branching coral: Fig. S13C). For seven types of substrate architectural characteristics, groupers showed significant positive and negative associations with overhang by coarse structure and flat, respectively (Tables S5–S8). However, for 25 substrate types, a significant positive associations with non-acroporid branching corals was found (Tables S6, S8). In contrast, significant negative associations with branching Acropora, bottlebrush Acropora and coral rubble were found (Tables S6, S8).

Butterflyfishes were primarily associated with large inter-branch space (staghorn Acropora), and some individuals were also associated with eave-like space (corymbose Acropora and tabular Acropora), overhang by fine branching structure (branching Acropora) and overhang by coarse structure (rock: Fig. S13D). Butterflyfishes showed a significant positive association with large inter-branch space (staghorn Acropora), whereas a significant negative association with overhang by fine branching structure (bottlebrush Acropora) (Tables S5–S8). A significant negative association with massive coral was also found (Tables S6, S8).

Overall trend of substrate association including the seventeen fish species

For the seven types of substrate architectural characteristics, PCA revealed that three architectural characteristics (eave-like space, large inter-branch space and overhang by coarse structure) showed major contributions for nocturnal fish associations (Fig. 10A). Cluster analysis revealed the 17 species could be divided into six groups (Fig. 10B, Fig. S14A). Two species (Scarus ghobban and Naso unicornis: group B), one species (Chaetodon lunulatus: group D) and five species (Scarus forsteni, S. niger, S. oviceps, S. rivulatus and S. schlegeli: group F) showed greater proportions in association with overhang by coarse structure, large inter-branch space and eave-like space, respectively. Other fishes belonging to three groups (group A, C and E) did not show greater proportion in association with any particular architectural characteristics. For fish size difference, four species (Ch. microrhinos, H. longiceps, S. niger and N. unicornis) showed relatively clear variations in substrate associations among difference size classes (Fig. S15). For the two species (Ch. microrhinos and H. longiceps), the main associated substrates changed from large inter-branch space to overhang by coarse structure as fish size increased (Figs. S15B, S15D). In contrast, the other two species (S. niger and N. unicornis) showed that the main associated substrates changed from eave-like space to large inter-branch space (Fig. S15G) and from eave-like space to overhang by coarse structure as fish size increased (Fig. S15K), respectively.

For 25 substrate types, PCA revealed that three substrate types (tabular Acropora, staghorn Acropora and rock) showed major contributions for nocturnal fish associations (Fig. 10C). Cluster analysis revealed 17 species could be divided into eight groups (Fig. 10D, Fig. S14B). Naso unicornis (group A), Chaetodon lunulatus (group D), Scarus schlegeli (group F) and two species (Scarus oviceps and S. rivulatus: group H) showed greater proportions in association with rock, staghorn Acropora, corymbose Acropora and tabular Acropora, respectively. Other fishes belonging to four groups (group B, C, E, G) and one species (Chaetodon trifascialis: group D) did not show greater proportions in association with any particular substrate type. For fish size difference, two species (Ch. microrhinos and H. longiceps,) showed that the main associated substrates changed from staghorn Acropora to rock as fish size increased (Figs. S16B, S16D). Two species (S. niger and N. unicornis) showed that the main associated substrates changed from tabular Acropora to staghorn Acropora (Fig. S16G) and from dead tabular Acropora to rock as fish size increased (Fig. S16K: dead tabular Acropora was shown as “other substrates” in Fig. S16K. See also Fig. S9), respectively.

Discussion

This study examined the nocturnal substrate association of 17 species from four fish groups, which was the first study in the North Pacific (Okinawan coral reef). The results of the present study could provide useful information as to what types of substrates should be protected and/or restored for fish habitat at nighttime as well as fishing locations for nighttime spear-fishing. It could also provide some guidance for the development and design of artificial reefs.

Parrotfishes

Most previous studies have conducted diurnal observations to clarify the spatial distribution in relation to topographic and substrate characteristics (Hoey & Bellwood, 2008; Hernández-Landa et al., 2014; Nanami, 2021) and foraging substrates (Nanami, 2016; Bonaldo & Rotjan, 2018). However, substrate associations for parrotfish species have not been sufficiently examined due to their highly diurnal activity (e.g., Welsh & Bellwood, 2012). Pickholtz et al. (2023) examined nocturnal substrate associations of seven parrotfish species in the Indian Ocean (Gulf of Aqaba), in which substrates were categorized into five types (branching coral, massive coral, soft coral, rock and artificial structure). In contrast, the present study conducted in the North Pacific (Okinawa) and categorized substrates into seven types in terms of architectural characteristics and 25 types in terms of more precise aspects (e.g., coral morphology, live coral or dead coral, and other non-coralline substrates).

Three species (Chlorurus microrhinos, C. spilurus and Hipposcarus longiceps) showed significant positive associations with large inter-branch space (staghorn Acropora). Pickholtz et al. (2023) revealed nocturnal substrate associations for three closely related species in the Indian Ocean (C. gibbus, C. sordidus and H. harid) and showed some individuals of the three species were associated with branching corals. These results suggest that substrates that were positively associated with parrotfishes are similar among closely related species.

Scarus ghobban and Chlorurus microrhinos showed significant positive associations with overhang by coarse structure. Nanami & Nishihira (2004) showed smaller-sized fish species (pomacentrids and juveniles of labrids of less than 10 cm in length) were associated with the base of massive corals as shelter due to their overhang structure. In contrast, Kerry & Bellwood (2012) suggested that massive corals showed less contribution for concealment of larger-sized fishes (over 10 cm in length), although a possibility that large massive corals might provide canopy effects by overhang at the base of the colony. The results of this study support this suggestion. Namely, overhangs provided by coarse structure serve to some degree as sleeping sites for larger-sized parrotfish individuals (TLs were 24 cm and over).

The remaining five species (Scarus forsteni, S. niger, S. oviceps, S. rivulatus and S. schlegeli) and C. spilurus showed significant positive associations with eave-like space (primarily provided by corymbose Acropora and tabular Acropora). As Kerry & Bellwood (2012) suggested, it was revealed that tabular corals provide concealment for some parrotfish species as sleeping sites due to their canopy structure.

Surgeonfishes

Naso unicornis and N. lituratus showed significant positive associations with overhang by coarse structure mainly provided by rock and eave-like space being mainly provided by tabular Acropora, respectively. Some N. unicornis were also associated with eave-like space provided by tabular Acropora. These findings suggest that canopy structure (overhangs and tabular structure) should be conserved as sleeping sites for these species.

Naso unicornis and N. lituratus are main fishery targets in coral reefs (Bejarano et al., 2013; Taylor et al., 2014) and nighttime spear fishing is a common method to catch inactive individuals of these species (Taylor et al., 2014). Conservation of critical substrates as sleeping sites could serve as fishing locations that can be utilized by fishermen.

Groupers

Plectropomus leopardus is diurnally active and nocturnally inactive (Matley, Heupel & Simpfendorfer, 2015). Broad-scale diurnal survey (several and several-tens of kilometer scale) have shown that a greater coverage of branching Acropora was positively related with greater density of this species (Nanami, 2021). In contrast, this species showed a significant positive association with eave-like space mainly provided by corymbose and tabular Acropora as sleeping sites. These results suggest that substrate types that affect the spatial distribution of the species may be different between daytime and nighttime. Plectropomus leopardus is a carnivore and its main prey items are small-sized fishes (St John, 1999). Since such small-sized fishes were often associated with branching Acropora, this species might occur at sites with greater coverage of branching Acropora for foraging during daytime but utilize eave-like space as sleeping sites during nighttime. Thus, multiple substrate types are needed to satisfy the ecological requirements of this species during both daytime and nighttime.

Diurnal observations revealed that large-sized Epinephelus ongus individuals (over 18 cm TL) showed a significant positive association with large inter-branch space that was created by staghorn Acropora (Nanami et al., 2013). In contrast, nocturnal observations by this study showed positive associations with overhang by coarse structure. Nanami et al. (2018) suggested that this species is nocturnally active since a greater home range size was observed at nighttime than daytime. This species might be associated with overhang by coarse structure for ambush foraging at nighttime.

Butterflyfishes

Chaetodon trifascialis showed positive associations with eave-like space (tabular Acropora) and large inter-branch space (staghorn Acropora). This species is an obligate coral polyp feeder and mainly feeds on polyp of tabular Acropora and corymbose Acropora (Pratchett, 2005; Nanami, 2020). This suggests that coral species providing large inter-branch space are important architectural structure as sleeping sites for this species, which was not indicated by diurnal observations for the clarifying foraging behavior. In contrast, tabular Acropora was also utilized as sleeping sites, suggesting that tabular Acropora is essential as both foraging and sleeping sites for this species.

Chaetodon lunulatus showed a significant positive association with large inter-branch space being provided by staghorn Acropora. In contrast, diurnal observations revealed that this species mainly feeds on polyps of encrusting, massive and non-acroporid corals, which do not provide large inter-branch space (Pratchett, 2005; Nargelkerken et al., 2009; Nanami, 2020). This indicates that C. lunulatus depends on staghorn Acropora as sleeping sites but it is not utilized as a foraging substrate, suggesting that various types of corals are essential for this species.

Chaetodon ephippium showed a significant positive association with large inter-branch space being provided by staghorn Acropora. In contrast, this species showed frequent bites on the surface of coral rubble, dead coral and rock (Nargelkerken et al., 2009; Nanami, 2020), probably due to catch invertebrates (Sano, Shimizu & Nose, 1984; Pratchett, 2005). This indicates that substrates utilization by C. ephippium was different between daytime and nighttime.

Chaetodon auriga did not show any significant associations with substrates. This species is facultative coral polyp feeder (Sano, Shimizu & Nose, 1984) and showed a greater number of bites on coral rubble and rocks (Nanami, 2020). Since this species was mainly associated with four types of substrate architectural characteristics (eave-like space, large inter-branch space, overhang by fine branching structure and overhang by coarse structure) but not associated with other three types of architectural characteristics (uneven surface, flat and macroalgae), this species utilized substrates with complex physical structure as sleeping sites. Since these four types of substrate architectural characteristics are provided by both live corals and rock, such substrates with greater complexity should be conserved as sleeping site for the species.

Overall, this study revealed large inter-branch spaces that created by staghorn Acropora was important physical structure as sleeping sites for the three species (C. trifascialis, C. lunulatus and C. ephippium) and substrates with complex physical structure were also important as sleeping site for C. auriga, which have not been revealed by diurnal observations in previous studies.

Variations in substrate association among different fish size classes

Four species showed clear variations in nocturnal substrate associations among different size classes. The two species (Ch. microrhinos and H. longiceps,) and one species (N. unicornis) showed that their main associated substrates changed from large inter-branch space (staghorn Acropora) to overhang by coarse structure (rock), and from eave-like space (dead tabular Acropora) to overhang by coarse structure (rock) as fish size increased, respectively. These results suggest that smaller- and larger-sized individuals were respectively associated with fine and coarse habitat structures, and various types of substrate architectural characteristics are needed for the various sizes of the three species as nocturnal sleeping sites. In contrast, S. niger showed that the main associated substrates changed from eave-like space (mainly tabular Acropora) to large inter-branch space (staghorn Acropora) as fish size increased, suggesting that various types of acroporid corals are needed for the various sizes of the species as nocturnal sleeping sites.

Implication about coral community degradation induced by climate change

Numerous studies have shown that coral species belonging to the genus Acropora is highly susceptible to coral bleaching by climate change (e.g., Marshall & Baird, 2000; Loya et al., 2001; McClanahan et al., 2004) and such degradation of the acroporid coral community causes significant declines of fish populations in coral reefs (Pratchett et al., 2008). All 17 species were nocturnally associated with acroporid coral, although the degree of association was species-specific. Especially, five species (Scarus oviceps, S. rivulatus, S. schlegeli, Chaetodon lunulatus and C. trifascialis) showed a greater proportion in association with acroporid corals. Some other species (Chlorurus microrhinos, C. spilurus, Hipposcarus longiceps, S. forsteni, S. niger, Naso lituratus, Plectropomus leopardus, Chaetodon ephippium) also showed positive associations with acroporid corals to some extent. In contrast, almost all fish species (except for one individual of P. leopardus) showed no associations with uneven structure without large space or overhang, flat and macroalgae, indicating fish avoidance of the three substrate architectural structural categories. These results suggest that the effects on coral degradation would negatively impact on the availability of sleeping sites for some fish species. This degradation would also cause a decline in fishing grounds for night spear fishing.

Conclusions

This study revealed nocturnal substrate associations of four coral reef fish groups (parrotfishes, surgeonfishes, groupers and butterflyfishes). In particular, the four fish groups were primarily associated with three architectural characteristics (eave-like space, large inter-branch space and overhang by coarse structure) that were primarily provided by tabular and corymbose Acropora, staghorn Acropora, and rock, which have not been revealed by diurnal observations in previous studies. These new insights will provide useful ecological information for effective conservation of biodiversity and ecosystem services of coral reef fishes. In particular, the death of acroporid corals caused by coral bleaching would decrease the availability of sleeping sites for some fish species. Consequently, it could lead to population declines of these fish species. Consideration of fish nocturnal substrate associations could provide more effective strategies for conservation and restoration of coral assemblages.

Supplemental Information

Supplemental Information 1 Relative frequency (%) of fish individuals associated with substrates and substrate availability for three size classes of Chlorurus microrhino s

Numbers above bars represent the number of individuals on the focal substrate. Black arrows represent significant positive association for the substrates that were examined by resource selection ratio (Manly et al., 2002: see Materials and Methods). Fish photograph was taken by the author (A. Nanami).

Supplemental Information 2 Relative frequency (%) of fish individuals associated with substrates and substrate availability for two size classes of Chlorurus spilurus

Numbers above bars represent the number of individuals on the focal substrate. Black and white arrows represent significant positive and negative associations for the substrates that were examined by resource selection ratio (Manly et al., 2002: see Materials and Methods). Fish photograph was taken by the author (A. Nanami).

Supplemental Information 3 Relative frequency (%) of fish individuals associated with substrates and substrate availability for three size classes of Hiposcarus longiceps

Numbers above bars represent the number of individuals on the focal substrate. Black arrows represent significant positive association for the substrates that were examined by resource selection ratio (Manly et al., 2002: see Materials and Methods). Fish photograph was taken by the author (A. Nanami).

Supplemental Information 4 Relative frequency (%) of fish individuals associated with substrates and substrate availability for three size classes of Scarus ghobban

Numbers above bars represent the number of individuals on the focal substrate. Fish photograph was taken by the author (A. Nanami).

Supplemental Information 5 Relative frequency (%) of fish individuals associated with substrates and substrate availability for three size classes of Scarus forsteni

Numbers above bars represent the number of individuals on the focal substrate. Black arrows represent significant positive association for the substrates that were examined by resource selection ratio (Manly et al., 2002: see Materials and Methods). Fish photograph was taken by the author (A. Nanami).

Supplemental Information 6 Relative frequency (%) of fish individuals associated with substrates and substrate availability for two size classes of Scarus niger

Numbers above bars represent the number of individuals on the focal substrate. Black arrows represent significant positive association for the substrates that were examined by resource selection ratio (Manly et al., 2002: see Materials and Methods). Fish photograph was taken by the author (A. Nanami).

Supplemental Information 7 Relative frequency (%) of fish individuals associated with substrates and substrate availability for two size classes of Scarus oviceps

Numbers above bars represent the number of individuals on the focal substrate. Black arrows represent significant positive association for the substrates that were examined by resource selection ratio (Manly et al., 2002: see Materials and Methods). Fish photograph was taken by the author (A. Nanami).

Supplemental Information 8 Relative frequency (%) of fish individuals associated with substrates and substrate availability for two size classes of Scarus rivulatus

Numbers above bars represent the number of individuals on the focal substrate. Black arrows represent significant positive association for the substrates that were examined by resource selection ratio (Manly et al., 2002: see Materials and Methods). Fish photograph was taken by the author (A. Nanami).

Supplemental Information 9 Relative frequency (%) of fish individuals associated with substrates and substrate availability for three size classes of Naso unicornis

Numbers above bars represent the number of individuals on the focal substrate. Black and white arrows represent significant positive and negative association for the substrates that were examined by resource selection ratio (Manly et al., 2002: see Materials and Methods). Fish photograph was taken by the author (A. Nanami).

Supplemental Information 10 Relative frequency (%) of fish individuals associated with substrates and substrate availability for two size classes of Naso lituratus

Numbers above bars represent the number of individuals on the focal substrate. Black arrows represent significant positive association for the substrates that were examined by resource selection ratio (Manly et al., 2002: see Materials and Methods). Fish photograph was taken by the author (A. Nanami).

Supplemental Information 11 Relative frequency (%) of fish individuals associated with substrates and substrate availability for three size classes of Plectropomus leoparudus

Numbers above bars represent the number of individuals on the focal substrate. Black arrow represents significant positive association for the substrates that were examined by resource selection ratio (Manly et al., 2002: see Materials and Methods). Fish photograph was taken by the author (A. Nanami).

Supplemental Information 12 Relative frequency (%) of fish individuals associated with substrates and substrate availability for two size classes of Epinephelus ongus

Numbers above bars represent the number of individuals on the focal substrate. Black arrow represents significant positive association for the substrates that were examined by resource selection ratio (Manly et al., 2002: see Materials and Methods). Fish photograph was taken by the author (A. Nanami).

Supplemental Information 13 Relative frequency (%) of fish individuals associated with substrates and substrate availability for the four families (parrotfishes, surgeonfishes, groupers and butterflyfishes)

Left figures represent results using the seven types of substrate architectural characteristics. Right figures represent results using 19 substrate types for parrotfishes, surgeonfishes and butterflyfishes, and 24 substrate types for groupers, respectively. Numbers adjacent to bars represent the number of individuals that were associated with the focal substrate. All fish illustrations were drawn by the author (A. Nanami).

Supplemental Information 14 Dendrogram of hierarchical clusters representing the nocturnal substrate associations of the 17 fish species via the group-average-linkage method using the Bray-Curtis similarity index in terms of five types of substrate architectural characteristics (A)

In (B), “Other substrates” includes 11 substrate types (bottlebrush Acropora, non-acroporid branching coral, foliose coral, Pocillopora, dead corymbose Acropora, dead tabular Acropora, dead staghorn Acropora, dead branching Acropora, dead non-acroporid branching coral, dead Pocillopora and coral rubble). Two types of substrate architectural characteristics (uneven structure and macroalgae) and seven substrate types (other coral, dead bottlebrush Acropora, dead foliose coral, dead other coral, soft coral, sand and macroalgae) are not included in analyses, since no fish individuals were associated with the substrates. For details about data, see ” Fig. S14 raw data.xls.”

Supplemental Information 15 Results of principal component analysis (PCA) for substrate association of three size classes of fishes based on five types of substrate architectural characteristics (Small: fish total length ≤ 29 cm; Medium: fish total length = 30 cm –39 cm; Larg

Since total length of five species (Scarus schlegeli, Chaetodon trifascialis, C. lunulatus, C. ephippium and C. auriga) were 29 cm or less for all individuals, only results for the smaller-sized individuals are shown for the species. Pie charts in (B-R) represent proportion of nocturnal substrate association for each size class for each species.

Supplemental Information 16 Results of principal component analysis (PCA) for substrate association of three size classes of fishes based on 18 substrate types (Small: fish total length ≤ 29 cm; Medium: fish total length = 30 cm –39 cm; Large: fish total length ≥ 40 cm)

Since total length of five species (Scarus schlegeli, Chaetodon trifascialis, C. lunulatus, C. ephippium and C. auriga) were 29 cm or less for all individuals, only results for the smaller-sized individuals are shown for the species. Pie charts in (B-R) represent proportion of nocturnal substrate association for each size class for each species.

Supplemental Information 17 Results of standardized selection ratio of the nine parrotfish species calculated by resource selection ratio (Table 3) for seven types of substrate architectural characteristics

Significant positive associations are shown as bold characters. N.S.: non significant associations. -: no fishes were found on the substrates.

Supplemental Information 18 Results of standardized selection ratio of the nine parrotfish species calculated by resource selection ratio (Table 4) for 25 substrate types

Significant positive associations are shown as bold characters. N.S.: non significant associations. -: no fishes were found on the substrates.

Supplemental Information 19 Results of standardized selection ratio of the two surgeonfish, two grouper and four butterflyfish species calculated by resource selection ratio (Table 5) for seven types of substrate architectural characteristics

Significant positive associations are shown as bold characters. N.S.: non significant associations. -: no fishes were found on the substrates.

Supplemental Information 20 Results of standardized selection ratio of the two surgeonfish, two grouper and four butterflyfish species calculated by resource selection ratio (Table 5) for 25 substrate types

Significant positive associations are shown as bold characters. N.S.: non significant associations. -: no fishes were found on the substrates.

Supplemental Information 21 Results of statistical significance of substrate association of the four families (parrotfishes, surgeonfishes, groupers and butterflyfishes) calculated by resource selection ratio for seven types of substrate architectural characteristics

Significant positive associations are shown as bold characters. N.S.: non significant associations. -: no fishes were found on the substrates.

Supplemental Information 22 Results of statistical significance of substrate association of the four families (parrotfishes, surgeonfishes, groupers and butterflyfishes) calculated by resource selection ratio for 25 substrate types

Significant positive associations are shown as bold characters. N.S.: non significant associations. -: no fishes were found on the substrates.

Supplemental Information 23 Results of standardized selection ratio of the four families (parrotfishes, surgeonfishes, groupers and butterflyfishes) calculated by resource selection ratio for seven types of substrate architectural characteristics (Table S5)

Significant positive associations are shown as bold characters. N.S.: non significant associations. -: no fishes were found on the substrates.

Supplemental Information 24 Results of standardized selection ratio of the four families (parrotfishes, surgeonfishes, groupers and butterflyfishes) calculated by resource selection ratio for 25 substrate types (Table S6)

Significant positive associations are shown as bold characters. N.S.: non significant associations. -: no fishes were found on the substrates.

Supplemental Information 25 Fig. 4 raw data

Supplemental Information 26 Fig. 5 raw data

Supplemental Information 27 Fig. 6 raw data

Supplemental Information 28 Fig. 7 raw data

Supplemental Information 29 Fig. 8 raw data

Supplemental Information 30 Fig. 9 raw data

Supplemental Information 31 Fig. 10 raw data

Supplemental Information 32 Table 3 raw data

Supplemental Information 33 Table 4 raw data

Supplemental Information 34 Table 5 raw data

Supplemental Information 35 Table 6 raw data

Supplemental Information 36 Substrate avaiability raw data

Supplemental Information 37 Fig. S1 raw data

Supplemental Information 38 Fig. S2 raw data

Supplemental Information 39 Fig. S3 raw data

Supplemental Information 40 Fig. S4 raw data

Supplemental Information 41 Fig. S5 raw data

Supplemental Information 42 Fig. S6 raw data

Supplemental Information 43 Fig. S7 raw data

Supplemental Information 44 Fig. S8 raw data

Supplemental Information 45 Fig. S9 raw data

Supplemental Information 46 Fig. S10 raw data

Supplemental Information 47 Fig. S11 raw data

Supplemental Information 48 Fig. S12 raw data

Supplemental Information 49 Fig. S13 raw data

Supplemental Information 50 Fig. S14 raw data

Supplemental Information 51 Fig. S15 raw data

Supplemental Information 52 Fig. S16 raw data

Supplemental Information 53 Raw data of Table S1.

Supplemental Information 54 Raw data of Table S2.

Supplemental Information 55 Raw data of Table S3.

Supplemental Information 56 Raw data of Table S4.

Supplemental Information 57 Raw data of Table S5.

Supplemental Information 58 Raw data of Table S6.

Supplemental Information 59 Raw data of Table S7.

Supplemental Information 60 Raw data of Table S8.

I express my grateful thanks to Masato Sunagawa and Sho Sunagawa for their assistance in the field, and the staff of Yaeyama Field Station, Japan Fisheries Research and Education Agency for support during the present study. Constructive comments from Magnus Johnson, Chris Norman, Andy Richardson and an anonymous reviewer were much appreciated. The present study complies with current laws in Japan.

Additional Information and Declarations

Competing Interests

Author Contributions

Animal Ethics

Data Availability

The author declares that they have no competing interests.

Atsushi Nanami conceived and designed the experiments, performed the experiments, analyzed the data, prepared figures and/or tables, authored or reviewed drafts of the article, and approved the final draft.

The following information was supplied relating to ethical approvals (i.e., approving body and any reference numbers):

The present study conducted fish sampling by spear gun, which is officially permitted by Okinawa Prefectural Government (fisheries coordinate regulation No. 37). This regulation shows that scientists can capture marine organisms with no restrictions about fishing gear for scientific purpose.

In Okinawan region, permissions about field observations are not required. This is because above-mentioned fisheries coordinate regulation does not require permission about field observations from Okinawa Prefectural Government.

The following information was supplied regarding data availability:

The raw measurements and data are available in the Supplementary Files.

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
