# Peer review of "Nocturnal substrate association of four coral reef fish groups (parrotfishes, surgeonfishes, groupers and butterflyfishes) in relation to substrate architectural characteristics"

_PeerJ, doi:10.7717/peerj.17772_

## Round 0.1 · original submission · Major Revisions

My apologies for the length of time it has taken to come to a decision. As you can see two reviewers have given detailed opinions but differ in their suggested outcomes. Particular criticisms are around the need for euthanizing the fish, the apparent discrepancy between collecting fish data and substrate data, the fact that the TL data do not appear to have been used, the fact that the figures cannot be read and some comments on the statistical techniques applied and the need for more clarity there. I would like to give you the opportunity to address the concerns of the reviewers before sending the paper out for a 2nd round of reviews.

·

Basic reporting

Clear, professional English language used throughout, but some statements require clarification.
• 89-91 Microhabitat-level substrate association – This statement is surprising considering the volume of papers already in the literature about habitat associations of several of the reef fish groups, particularly those of commercial fisheries value. The nuance of the inclusion of “microhabitat-level” might be an attempted qualifier to exempt this statement, but there is no previous definition of a “microhabitat”. This term needs to be defined in the text.
Literature generally well referenced & relevant. However, greater use could be made of several resources.
• 72-83 Ecosystem services – this is substantially oversimplified and overlooks key ecosystem services provided by coral reef fish. Given the ecological focus of this manuscript, and the citation of ecosystem service papers such as Woodhead et al (2019), the scope and value of the ecosystem services provided by coral reef fish species could be explored much more effectively than currently presented in the manuscript.
Figures are relevant, however the quality of Figures 4 and 5 is insufficient, and major reorganisation is required.
Raw data are supplied in Table form for fish surveys (Table 1), but not for raw substrate data taken from video stills – this does not adhere to PeerJ standards.

Experimental design

At face, the submission does appear to present original, primary research. However, the research question is not placed strongly into context of the importance of the resulting knowledge, particularly with regard to nocturnal habitat associations. It is unclear as to why this information is particularly important to management or value of ecosystem services, such as fisheries or biodiversity and the author does not make a strong case to support the research in this.

Investigation has not been performed to a high technical & ethical standard
• 133-136 (also 120-121) Methodology – The requirement for destructive sampling/euthanasia is felt to be contentious in this instance. Although the author states that the activity (spearing of partially-visible individuals) is permitted under the local government fisheries regulations, the ethics of doing so within the scope of this project, in a no-take reserve, are less convincing. Invasive sampling might well be justifiable in other studies, such as diet analysis requiring stomach content analysis (such as Nicholson, M.D., Pagán, J.A., Hendrick, G.C. and Sikkel, P.C., 2024. Functional diversity among coral reef fishes as consumers of ectoparasites. Coral Reefs, pp.1-11). However, for the purpose of establishing total length this invasive methodology is considered excessively destructive. Given the information provided in the methodology (line 132), the total length of unconcealed individuals appears to have been estimated by visual means. Given this margin for error naturally introduced by observer estimation of TL, could the length estimation not also be made from partially-concealed individuals using morphometric and anatomical distance estimates (either in-situ or using photography)? Similar non-invasive approaches were used by a study cited in the current manuscript (Pickholtz et al. 2023) but not adopted by the author in their experimental design.
As total length (TL) data was not apparently used in any reported analysis, the use of a euthanasia-based method for collection of this data specifically for this study is not considered defensible.

• 144-149 Methodology (substrate) – It is confusing why substrate availability data was collected at whole-site level, when finer scale information was available during the fish surveys. Considering that the study discusses the concept of “micro-habitats”, whole-site classification of habitats is inappropriate. Additionally, as the fish surveys were not conducted using fixed transects (or similar), using instead random zigzag patterns, the likelihood of fish positions being represented in subsequent habitat surveys is very low. A more appropriate and replicable methodology would have been to collect habitat data in-situ during the fish surveys, when the first SCUBA diver had already identified when fish were associated with substrates.

• 151-161 Substrate classification - Substrate classification methodology should be reported in the manuscript, the author refers to a previous work (Nanami, 2020) which merely then refers to repeatedly preceding studies by the same author (2018, 2013). The robustness of the substrate classification would be greatly improved by establishing the location-specific level for substrate (habitat) classification using a hierarchical clustering method. A useful publication for an introduction to this is Northrup, J.M., Vander Wal, E., Bonar, M., Fieberg, J., Laforge, M.P., Leclerc, M., Prokopenko, C.M. and Gerber, B.D., 2022. Conceptual and methodological advances in habitat‐selection modeling: guidelines for ecology and evolution. Ecological Applications, 32(1), p.e02470. Without this classification process, distinctions between the substrate types (Table 2) appear arbitrary. For example, “coral rubble” (line 160) may have similarity with comparable substrates in terms of structural complexity and interstitial spaces. Instead, classification into substrate types is assumed to be on a visual basis, rather than measurable similarity or dissimilarity of structure. Consequently, “coral rubble” is grouped in to “(6) flat” (line 173) substrate type, which seems counterintuitive. This kind of clustering for substrate-level groupings would be an ideal accompaniment to the dendrograms presented in the supplementary materials.

Validity of the findings

The rationale and benefit of the presented findings to the literature are uncertain.
• 297-298 Discussion – related to previous comment, the term “nocturnal” is inappropriate as habitat information was not collected during the hours of darkness (lines 144-149). Furthermore “microhabitat-level” is untrue, not least as the term “microhabitat” has not been defined. It is genuinely puzzling why the actual habitat that the fish were found in was not recorded in-situ (also as Figure 2 demonstrates that photography was an option for consequent ex-situ interpretation of substrate information). Without this in-situ and contemporary measurement, there is no justification for claims that this study analyses the nocturnal substrate association of fish species, as there is no accounting for any diurnal variability in substrate association. For example in lines 312-313, the author reports Chlorurus microrhinos showed association with large inter-branch space substrate – but this is based on daytime habitat surveys and nocturnal fish surveys - so while it may show that C. microrhinos was present in habitats with large inter-branch space substrate in general, any claims that this is specifically a nocturnal association are inaccurate. Consequently, the data collected in this study are not sufficient or appropriate for the stated study objective (lines 111-113). This is possibly indicative of either an inadequate methodology for the planned objective, or testing the wrong hypothesis on data already collected.
Conclusions are clearly stated (lines 409-410) but are not rigorously supported by the data collected, due to deficiencies in the experimental design.
• 415-419 concluding statement – the value of the research to management and conservation is not convincing, due to aforementioned data issues and a stretched interpretation of the findings.
• 418 poor grammar/confusing language – “by fishes will enable to propose more”

Additional comments

The aims of the study were ambitious, and the use of resource selection ratio and PCA analysis is appreciated. However, there are significant deficiencies in the experimental design that likely resulted in data that were inadequate to test the stated hypothesis. Additionally, unnecessary data collection is reported in the methods section, such as total length (with associated euthanasia – see specific comment), which has no further bearing on the analysis or hypotheses being tested.
Suggested improvement would be to remove the “nocturnal” and “microhabitat” aspects of the narrative. The habitat associations that the author has presented are demonstrable without these additional nuances, and may have greater validity without them. By removing these specifics from the stated objectives and interpretations, and also increasing the commentary and direct application to local conservation/management, the value of this study could be increased considerably.

Reviewer 2 ·

Basic reporting

Figures 4 and 5 are completely unreadable in the pdf form. If this is not a technical issue, these figures need to be cut in half and put half of the species on the right so the whole image can be enlarged. I was able to view the figures through the supplied png’s but ensure the figure is readable at typical vertical space allowed (~ 234mm).

Experimental design

Ln 212. There is no mention of the actual statistical analysis done or what software/packages was used to conduct the analysis.

Ln 192-193. “Substrates without any association with fish were excluded from the analysis.” The active avoidance of specific architect or habitat characteristic may be just as interesting and informative. For example, no parrotfish in the current study used flat, uneven or macroalgae as nocturnal habitat. This algins with your direction of conservation of some features (over others), as if reefs were to degrade into less complex and algae state (flat and macroalgae), fishes would have no sleeping habitat. These should still provide selection ratio below 1 and would likely be significant, showing avoidance. Can you please explain why these were removed for some species and provide a sentence in the text to explain this.

Ln 179-181. The selection ratio used is synonyms for Ivlev’s forage ratio, which is not the best selective index to use, Vanderploeg and Scavia's Relativized Electivity, E* is (See Lechowicz 1982). Can you please provide justification on why you used the selection ratio?
• Lechowicz, M. J. (1982). The sampling characteristics of electivity indices. Oecologia, 52, 22-30.

Validity of the findings

no comment

Additional comments

This manuscript describes a field study designed to quantify selectivity of nocturnal substrate associations in four coral reef fish groups (parrotfishes, surgeonfishes, groupers and butterflyfishes). The authors demonstrate that some species exhibit strong associations with particular architectural characteristics. Overall, I think it a well-planned out study that addresses an important gap in our knowledge of habitat use during a vulnerable period (nocturnal) for coral reef fishes. There are, however, several things that I would like to clarify to better appreciate the paper. The methods and statistics used were appropriate for answering the research question, however I feel they need to be more thoroughly described and justified (see specific comments below). Additionally, I believe the author is missing an important analysis to fully cover the presented aims (i.e. family level analysis).




Ln 111. The presented aim is “to clarify nocturnal substrate associations of four coral reef fish groups (parrotfishes, surgeonfishes, groupers and butterflyfishes)..” I feel that running a family level analysis on the selectivity of architectural characteristics and substrates prior to species level would provide a more succinct and broader message through the paper. This would also allow the author to include more data from Table 1, for species < 10. Additionally, this could provide interesting comparisons for particular species which don’t follow the family level norm.

Ln 127. Is there any mention of the size of the area surveyed in each dive or site. Or was this just a time-based survey?

Ln 70. Species-specific habitat associations to specific substrates or structural complexities have recently been shown to also shown to influence populations through survivorship (See Fakan et al. 2024).
• Fakan, E. P., McCormick, M. I., Jones, G. P., & Hoey, A. S. (2024). Habitat and morphological characteristics affect juvenile mortality in five coral reef damselfishes. Coral Reefs, 1-13.

Ln 151-161. This whole section should be removed. Everything listed here is already in Table 2, which is referenced. I suggest you combine “Substrate categorization” and “Definition of substrate architectural characterises” together and remove the list of 25 categorizes from the text to improve clarity and save space.

Figure3 is very nicely put together! The pictures help to visualize and the green marking helps to define the architectural characteristics which were assessed.

Ln 163-173. Similar to an earlier comment there is unnecessary lists of habitat characteristics. The figure is referenced, and the figure has “corymbose Acropora, tabular Acropora, foliose coral …” already in it; which is also already listed in Table 1. For clarity and space, I suggest removing all habitat characteristics and just leave “ … seven types as follows (Table 2, Fig. 3): (1) eave-like space, (2) large inter-branch space, (3) overhang provided by protrusion of fine branching structure…” .

Ln 179-181. The selection ratio used is synonyms for Ivlev’s forage ratio, which is not the best selective index to use, Vanderploeg and Scavia's Relativized Electivity, E* is (See Lechowicz 1982). Can you please provide justification on why you used the selection ratio?
• Lechowicz, M. J. (1982). The sampling characteristics of electivity indices. Oecologia, 52, 22-30.

A useful way of presenting the selection ratios used in the current study is to standardize them so they add to 1, which is called Manly’s standardized selection ratio, See equation 4.10 and table 4.1 pg 51-52 of Manly et al. This index provides you with a “B” which can easily be interpreted ecologically, eg a B = 0.326 for Eave-like would mean this species is > 3 times more likely to select for eave-like habitats. You may consider adding a B table along with resource selection ratio.
• Manly, B. F. L., McDonald, L., Thomas, D. L., McDonald, T. L., & Erickson, W. P. (2007). Resource selection by animals: statistical design and analysis for field studies. Springer Science & Business Media.

Ln 192-193. “Substrates without any association with fish were excluded from the analysis.” The active avoidance of specific architect or habitat characteristic may be just as interesting and informative. For example, no parrotfish in the current study used flat, uneven or macroalgae as nocturnal habitat. This algins with your direction of conservation of some features (over others), as if reefs were to degrade into less complex and algae state (flat and macroalgae), fishes would have no sleeping habitat. These should still provide selection ratio below 1 and would likely be significant, showing avoidance. Can you please explain why these were removed for some species and provide a sentence in the text to explain this.


Ln 195-204. It is unclear what this data modification is for. Please explicitly say what it is doing. It seems to be controlling for the substrate proportion at each site by the time sampling at those particular sites. More detail is needed in the text about this.


Figures 4 and 5 are completely unreadable in the pdf form. If this is not a technical issue, these figures need to be cut in half and put half of the species on the right so the whole image can be enlarged. I was able to view the figures through the supplied png’s but ensure the figure is readable at typical vertical space allowed (~ 234mm).

Ln 212. There is no mention of the actual statistical analysis done or what software/packages was used to conduct the analysis.

Ln 334-341. This paragraph is very repetitive. Of the six sentences, three basically say X or Y habitat feature should be conserved for species Z. I suggest combining these as protecting features X and Y would be most beneficial to conserves surgeon fishes or something of than nature.

Ln 342-345. You mention that nighttime fishing is used to catch some of these species. Was this mentioned in the introduction? This seems like a point worth making both in the introduction and at the very beginning of the discussion, to remind the reader why preserving ideal resting spot is important and could have positive ecological outcomes.

Ln 360. Mentions E. ongus individuals >18cm showed an association with staghorn Acropora. Table 1 has a minimum TL for this species at 10cm. Is this an error or was TL separate the species into size classes? In terms of the whole article, TL could be an interesting thing to consider explore (maybe at the family level). The size of the fish will physically determine in which spot a fish can rest. It would be interesting to see if some of these architectural characteristics are more useful at certain sizes and/or change as size increases. For example, if the aim is to preserve larger higher fitness individuals maybe the eave-like space is more useful than large inter-branching space.

Ln 387. You have not actually discussed the results for Chaetodon Auriga. This species did not select for any specific sites. I find this to be interesting and hypothesizes to explain this result should be discussed here. Maybe they are just less selective. Does it have to do with behaviour maybe they are ‘preferred’ spots are taken before they decided to rest. Is it related to body size? This species had the largest range in TL of the study. Maybe large and small individuals choose different habitats, which increased your variability.

---

## Round 0.2 · Major Revisions

I think that the major point made by the reviewer, that you have not used the length data still needs to be addressed. There are analyses you could carry out to see if there is any link between total length of fish and the type of refuge that they use. This could satisfy the justification for euthanising animals for research. I had a really quick look at size v habitat for parrotfish based on table 1 and there may be a preference for coarse overhang by larger fish. The alternative would be to remove all reference to fish length from the paper as it is not relevant to the study but this feels, to me, like a dishonest approach and you do discuss length and habitat in your discussion. I also think that the interaction of refuge preference and family may be important. I hope you will consider this approach. I have classed the work as needing major revisions because some re-analysis may be required but I don't think the work will be very time-consuming. I will consult with the managing editors upon resubmission and, given that the paper has been reviewed twice already, seek to avoid sending out to reviewers again if you do the size-based analysis.

A minor comment for line 113: "euthanased immediately to minimize suffering." might be better wording

·

Basic reporting

Author has satisfactorily addressed key issues identified in the original review (see comments and author responses)

Experimental design

Notable issues are still present with reference to comments/responses 1-8 and 1-9.

Reviewer Response 1-8: Lead author provides fair rationale for not using photography for length estimation. However, it is unclear from the original or revised methodology why Total Length (TL) data were required to be collected during surveys in the first place. Associated also with comment and response 1-9, the size range stated by the lead author appear to be irrelevant. If the study was restricted to studying, for example, only individuals which were between certain body lengths then the collection of the total length (TL) data (and possibly associated destructive sampling) might be justified. However, total length (TL) data are not used in the habitat association or selection ratio statistics, nor are they reported as products from the PCA. Therefore the question remains, why was it important or necessary to collect Total Length (TL) data at all? By extension, therefore, the case for destructive sampling to obtain these data is very weak.

Reviewer Response 1-9: As covered in 1-8, where is the analysis that actually uses the Total Length (TL) data? The manuscript reports the size ranges in Table 1, but this information is meaningless for the presented analyses and study hypothesis. Were the point of the study to co-investigate total length v substrate associations then the total length (TL) data would have value. Without this the Total Length (TL), size class information, and destructive sampling are superfluous.
Key recommendation would be to remove the size class information, and by extension the methodology content where Total Length (TL) is measured either in-situ or by euthanasia. Removing the references to destructive sampling would not impede the effectiveness or nature of the analysis presented, and also avoid questions about the relative ethics of the spearing methodology. Without the removal of this, and were the manuscript to be accepted by the journal, it is considered that further concerns would likely be raised by readers which impacts on the reputations of both the author and the journal.

Reviewer Response 1-10: Clarification of the methodology and finer details, such as the use of GPS, in the revised manuscript has helped to remove the ambiguity of the original methodology reporting.

Validity of the findings

Reviewer Response 1-13: The original methodology was unclear. Thus on original review, the methodology and subsequent analysis did not appear to answer the key hypothesis or research objective(s). However, the author has made numerous changes in response to multiple comments on the original material, notably the removal of reference to “microhabitats” and revision of the methodology to make clearer the link between the nocturnal survey data and wider daytime substrate survey. These changes have narrowed the focus of the study and make the conclusions much more applicable.

Reviewer Response 1-14: Additional comments on revised manuscript and author response attached to relevant comment 1-10.

Reviewer Response 1-15: There are still notable issues with aspects of the methodology, please refer to comments and responses 1-8 and 1-9

Additional comments

The efforts that the author has made to address the concerns from the original review are noted and appreciated. However, significant issues around aspects of the methodology remain (1-8 and 1-9).

---

## Round 0.3 · Minor Revisions

I have taken the opportunity to read through your manuscript again and make some suggestions for very minor edits, just for clarity of expression. Most of these can be implemented by simply accepting the tracked changes. I think the manuscript is now much improved. A minor point: there is a section where you cite only your own work. Although it is very relevant and appropriate, it is often regarded as good practice to supplement your own work with that of others - e.g. your point about Chaetodon feeding mainly on encrusting and massive coral polyps - this may be true in this location but others have suggested that elsewhere they feed on other things such as tabular Acropora.

---

## Round 0.4 · accepted · Accept

Thank you for engaging so well in the review process. I think you have responded to all of the comments made and I am happy with the latest version. I think it is ready for publication. The point about fish needing slightly different habitats by day and by night is particularly interesting, and also the fact that your work could be really useful in the design of artificial reefs for conservation or fisheries.